# Mi-2/NuRD complex protects stem cell progeny from mitogenic Notch signaling

**Evanthia Zacharioudaki, Julia Falo Sanjuan, Sarah Bray***

Department of Physiology, Development and Neuroscience, University of Cambridge, Cambridge, United Knigdom

**Abstract** To progress towards differentiation, progeny of stem cells need to extinguish expression of stem-cell maintenance genes. Failures in such mechanisms can drive tumorigenesis. In *Drosophila* neural stem cell (NSC) lineages, excessive Notch signalling results in supernumerary NSCs causing hyperplasia. However, onset of hyperplasia is considerably delayed implying there are mechanisms that resist the mitogenic signal. Monitoring the live expression of a Notch target gene, *E(spl)mγ*, revealed that normal attenuation is still initiated in the presence of excess Notch activity so that re-emergence of NSC properties occurs only in older progeny. Screening for factors responsible, we found that depletion of Mi-2/NuRD ATP remodeling complex dramatically enhanced Notch-induced hyperplasia. Under these conditions, *E(spl)mγ* was no longer extinguished in NSC progeny. We propose that Mi-2 is required for decommissioning stem-cell enhancers in their progeny, enabling the switch towards more differentiated fates and rendering them insensitive to mitogenic factors such as Notch.

DOI: https://doi.org/10.7554/eLife.41637.001

## Introduction

An important property of many stem cells is their ability to undergo asymmetrical divisions, generating one progeny cell that retains stem cell identity while the other is routed towards differentiation. Critically, the stem cell properties must be rapidly shut off in the latter to ensure it transitions to a different state. Failure to do so effectively can result in reactivation of stem cell programmes in the progeny, leading to hyperplasia and tumorigenesis. For example, a subset of the cells in many solid tumours (e.g glioblastomas, neuroblastomas) have characteristics of stem cells, deploying programmes that normal stem cells would use for tissue development and repair to support the development and progressive growth of the tumour (*Azzarelli et al., 2018*; *Lathia et al., 2015*; *Reya et al., 2001*; *Singh et al., 2004*). How some cells reacquire stem-cell properties and what events are required to ensure that daughter cells are normally programmed towards differentiation are currently unclear.

Neural Stem Cells, known as neuroblasts (NBs) in *Drosophila*, have served as a powerful model to identify mechanisms that regulate asymmetrical stem-cell divisions. Similar to mammalian neural stem cells, NBs divide asymmetrically to generate a larger daughter cell that retains stem-cell identity and a smaller daughter cell that will follow a more differentiated fate (*Homem et al., 2015*; *Knoblich, 2008*; *Sousa-Nunes et al., 2010*). After entering quiescence at the end of embryogenesis, NBs resume divisions in larval stages, upon receiving proper nutrition signals from their adjacent glial cells (*Chell and Brand, 2010*; *Sousa-Nunes et al., 2011*). In each division, key cell fate determinants such as Prospero, Numb and Brat are segregated into the smaller progeny cell. Together these factors help switch off the stem-cell programme and their loss disturbs the balance between self-renewal and differentiation (*Bello et al., 2006*; *Betschinger et al., 2006*; *Choksi et al., 2006*; *Hirata et al., 1995*; *Ikeshima-Kataoka et al., 1997*; *Knoblich et al., 1995*; *Lee et al., 2006b*; *Rhyu et al., 1994*). For example, Numb inhibits Notch activity, and the loss of Numb or other

*For correspondence:
sjb32@cam.ac.uk

Competing interests: The authors declare that no competing interests exist.

mechanisms giving excess Notch signalling can lead to hyperplasia (*Bowman et al., 2008*; *Lee et al., 2006a*; *Wang et al., 2006*; *Weng et al., 2010*). However, certain lineages, so-called Type I lineages, appear resistant to the ectopic Notch activity as the majority of their daughter cells continue to be routed towards differentiation even in these conditions.

Type I and Type II NBs can be distinguished based on their mode of division as well as the differential expression of specific molecular markers. In Type I lineages, the smaller daughter, so-called Ganglion Mother Cell (GMC), divides only once to generate two post-mitotic neurons or glia cells. In Type II lineages, the smaller progeny of the NBs become intermediate neural progenitors (INP), which then divide three to six times asymmetrically to self-renew and to generate GMCs that, like the GMCs derived from Type I NBs, then divide into two post-mitotic neurons or glia cells (*Bello et al., 2008*; *Boone and Doe, 2008*; *Bowman et al., 2008*; *Izergina et al., 2009*; *Kang and Reichert, 2015*; *Knoblich, 2008*). Both Type I and Type II neuroblasts exhibit high levels of Notch activity and express key targets *deadpan* (*dpn*), *Enhancer of split-HLHmγ* (*E(spl)mγ*) and *klumpfuss* (*klu*), that are shut off in the progeny (*Almeida and Bray, 2005*; *Berger et al., 2012*; *San-Juán and Baonza, 2011*; *Xiao et al., 2012*; *Zacharioudaki et al., 2016*; *Zacharioudaki et al., 2012*). Type II lineages are highly sensitive to excessive Notch activity, which brings about ectopic expression of stem cell markers in INPs to drive NB-like behaviour and tumour formation (*Bowman et al., 2008*; *Wang et al., 2006*; *Weng et al., 2010*; *Zacharioudaki et al., 2016*; *Zacharioudaki et al., 2012*). In contrast, Type I lineages only become hyperplastic under extreme conditions. Many factors have been found that enhance the sensitivity of Type II lineages to hyperplasia (*Bayraktar and Doe, 2013*; *Berger et al., 2012*; *Eroglu et al., 2014*; *Koe et al., 2014*; *Liu et al., 2017*; *Weng et al., 2010*; *Xiao et al., 2012*; *Xie et al., 2016*); however, few of these moderate the onset of hyperplasia in Type I lineages and it remains unclear how ultimately Type I progeny cells become engaged to re-initiate a stem-cell program.

Monitoring the emergence of hyperplasia in NB lineages with constitutive Notch activity, we show that prolonged exposure is required before NB-like tumour cells are formed in Type I lineages. Furthermore, these NB-like cells arise from a reversion of the post-mitotic progeny, rather than from newly born GMCs. Reasoning that there are intrinsic mechanisms that attenuate stem-cell programmes and Notch activity in NB progeny, we screened for genes whose depletion accelerated tumourigenesis of Type I NB lineages. To do this, we used RNAi knock-down and focused on genes that have been associated with tumour formation in human neuroblastomas. Our results reveal that Mi-2, and other members of the NuRD ATP-remodelling complex, suppress the emergence of stem-cell characteristics in Notch-induced conditions. We propose Mi-2/NuRD complex normally acts to decommission enhancers that are active in the stem cells, so that they become refractory to Notch and other signals in the stem-cell progeny.

## Results

### NB lineages are initially resistant to excessive Notch signalling

Prolonged expression of constitutively active Notch in NB lineages results in their overproliferation and the formation of many NB-like cells expressing key targets such as *dpn*, *E(spl)mγ* and *klu* (*Bowman et al., 2008*; *Wang et al., 2006*; *Weng et al., 2010*; *Xiao et al., 2012*; *Zacharioudaki et al., 2016*; *Zacharioudaki et al., 2012*). To better characterise the onset and progression of Notch-driven hyperplasia, constitutive active Notch (*NΔecd*, here referred to as Nact) was expressed for a short (8 hr), medium (24 hr) or long (48 hr) period of time in larval Type I NBs (via *grhNB-Gal4 Gal80ts*) to produce nerve cords bearing supernumerary Dpn-expressing cells (*Figure 1A,B*). In the early stages of Notch-driven hyperplasia (8 hr), only a small proportion of Type I lineages turned hyperplastic (22%, *Figure 1C*) and these contained only a few extra cells expressing stem-cell markers. Testing a panel of such stem-cell markers revealed that *E(spl)mγ* and *dpn* were the first to be expressed and, of the two, *E(spl)mγ* appeared to be the earliest as there was a subset of cells in hyperplastic lineages that express *E(spl)mγ-GFP* only (2 ± 0.2; *Figure 1A,D*; *Figure 1—figure supplement 1*). Slightly more cells per lineage expressed both *E(spl)mγ-GFP* and Dpn (6.6 ± 0.5; *Figure 1A,D*). Nevertheless, it is striking that relatively few Type I lineages exhibit ectopic expression of these direct Notch targets even after 8 hr of exposure to active Notch.

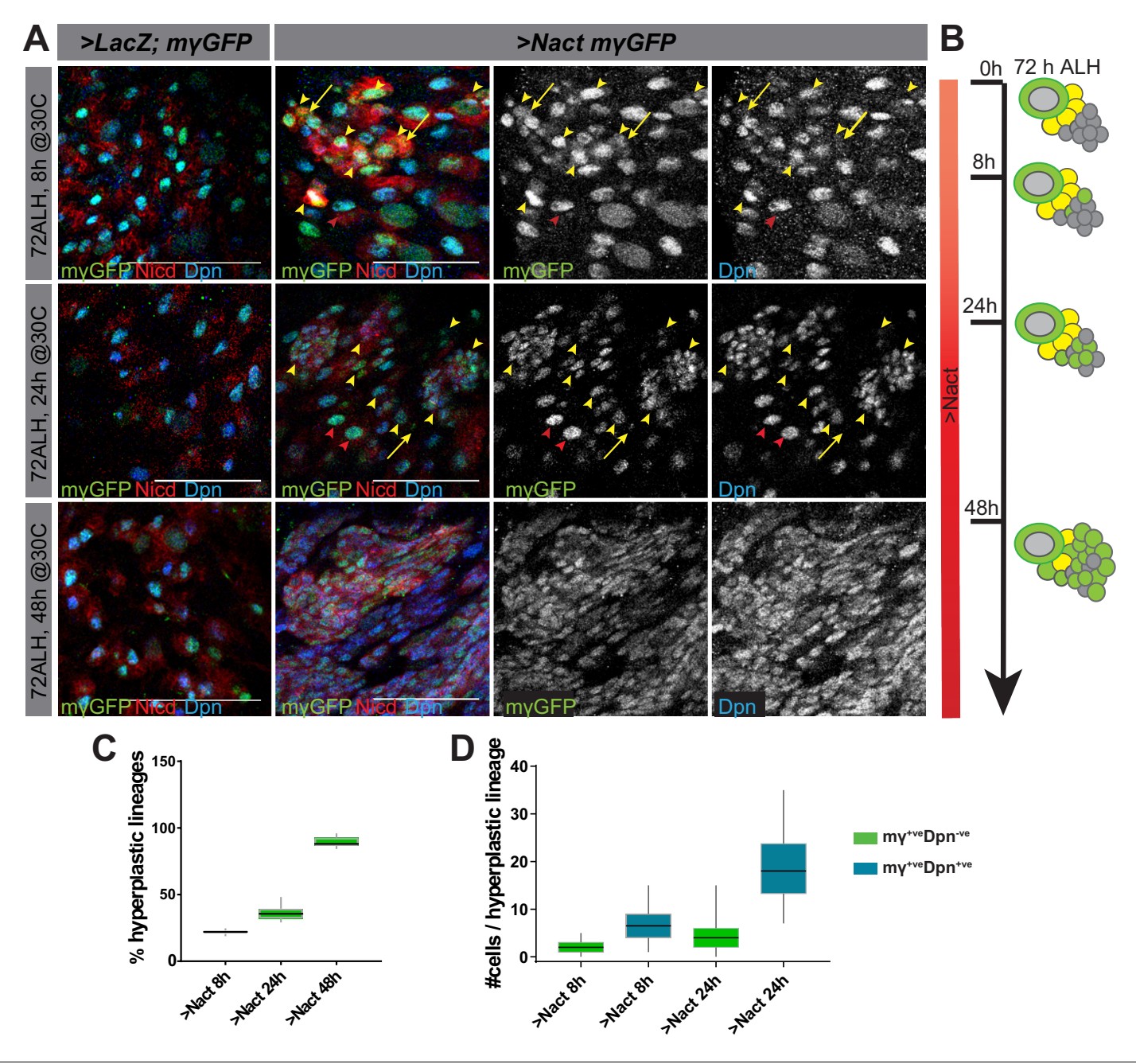

**Figure 1.** Delayed onset of hyperplasia in NB lineages expressing constitutively active Notch. (A) Expression of stem-cell markers in wild type (*grhNBGal4 >LacZ, E(spl)mγ-GFP*) ventral nerve cords (VNC) and in VNCs where NSC lineages were exposed to constitutive Notch activity (Nact; (*grhNBGal4 >NΔecd; E(spl)mγ-GFP*). Gal80ts was used to control the onset of expression to provide 8, 24 or 48 hr of Gal4. *E(spl)mγ-GFP* (green or white) and Dpn (blue or white) two Notch-responsive genes expressed in NSCs become upregulated in longer exposure times. High levels of *NΔecd* (anti-NICD, red) are present at even the earliest time-point. Red arrowheads indicate normal lineages, yellow arrowheads indicate hyperplastic lineages, yellow arrows indicate *E(spl)mγ-GFP*[+ve], *Dpn*[-ve] progeny. Scale bars: 25 μm. (B) Schematic representation of NB lineages at different times of Nact exposure; NBs, large green cells with grey nucleus, GMCs yellow and neurons grey. Ectopic NB-like cells are depicted as intermediate sized green cells. (C) Percent of lineages that were hyperplastic following 8 hr, 24 hr and 48 hr of Nact expression. Box represents IQR, black line indicates median and whiskers indicate ±1.5 × IQR. N = 15, three experiments. (D) Number of cells per hyperplastic lineage that are *E(spl)mγ-GFP*[+ve] Dpn[-ve] (green) or *E(spl)mγ-GFP*[+ve] Dpn[+ve] (blue) in 8 or 24 hr Nact expression. Box represents IQR, black line indicates median and whiskers indicate ±1.5 × IQR. N = 120, three experiments.

DOI: https://doi.org/10.7554/eLife.41637.002

The following figure supplements are available for figure 1:

*Figure 1 continued on next page*

*Figure 1 continued*

**Figure supplement 1.** E(spl)mγ and Dpn are among the earliest NB markers expressed by progeny of NB lineages exposed to constitutively active Notch.

DOI: https://doi.org/10.7554/eLife.41637.003

**Figure supplement 2.** Stem cells with constitutively active Notch divide asymmetrically.

DOI: https://doi.org/10.7554/eLife.41637.004

**Figure supplement 3.** The onset of hyperplasia in NB lineages expressing constitutively active Notch is delayed irrespectively of the age of the animal.

DOI: https://doi.org/10.7554/eLife.41637.005

Continued expression of Nact for 24 hr led to an increase in the number of Type I lineages with ectopic NB markers (36.3%; *Figure 1A,C*), as well as to an expanded number of cells per lineage with ectopic *E(spl)mγ-GFP* only (4.6 ± 0.6; *Figure 1A,D*) and with both *E(spl)mγ-GFP* and *dpn* expression (18.8 ± 1.1; *Figure 1A,D*). However, it was only with more prolonged Notch activity (48 hr) that the majority of lineages became hyperplastic (89.3%; *Figure 1A,C*) with a large fraction of the cells in each lineage expressing stem-cell markers so that large regions were occupied by NB-like cells (*Figure 1A*). Notably, the cells that acquired stem-cell characteristics were intermediate in size between a GMC and a NB, suggesting that they do not arise from a symmetrical division of a pre-existing NB. Furthermore, the NBs themselves continued to divide asymmetrically even in the presence of excessive Notch signaling (*Figure 1—figure supplement 2*). To rule out the possibility that the change in tumourigenic potential was due to the age of the NBs rather than the time of exposure to Notch activity, we also performed experiments where we varied the time of onset of exposure. This yielded identical results, that is the extent of hyperplasia correlated with the duration of exposure not the developmental stage at which the NBs were exposed (*Figure 1—figure supplement 3*).

In summary, even after 24 hr of exposure to Notch activity, only a few Type I stem-cell lineages become hyperplastic and these contain only a small number of cells expressing the 'early' stem-cell identity markers, *E(spl)mγ* and *dpn*. This differs from the situation in Type II lineages, which undergo much more severe overproliferation and suggests that there is a mechanism conferring resistance to Notch in the Type I NB progeny. With more prolonged Notch activity, this resistance is overcome and a large majority of cells acquire stem-cell like markers (*Figure 1B*).

## Hyperplasia involves NSC progeny re-acquiring stem-cell properties

Notch activity drives additional cells to acquire NB-like characteristics. However, it is unclear whether this arises at the time of NB division, so that the GMC retains a stem cell fate, or whether the neuronal progeny re-acquire stemness. To distinguish between these possibilities, we analysed the expression of *E(spl)mγ-GFP* in NB lineages in real time, by culturing NBs from normal brains and Notch-driven hyperplastic brains (after 24 hr with Nact). NBs were imaged continuously for 10 to 14 hr and their progeny tracked following each division to determine whether or not they maintained or reacquired *E(spl)mγ-GFP* expression. In these conditions, normal NBs underwent asymmetric cell divisions and *E(spl)mγ-GFP* expression was rapidly extinguished in the GMC progeny (*Figure 2A,C,E* and *Figure 2—video 1*). In the NB itself the levels fluctuated, with a defined temporal pattern through the cell cycle where the highest expression occurred as the NBs entered and exited mitosis (*Figure 2A,C,E* and *Figure 2—video 1*.

In the presence of high levels of Notch activity, the NBs continued to undergo asymmetric cell divisions and the expression of *E(spl)mγ-GFP* was extinguished in the GMC, albeit with slightly slower kinetics than in normal conditions (*Figure 2B,D,F* and *Figure 2—video 2*). After a few hours, however, some of the NB progeny reacquired *E(spl)mγ-GFP* expression. The reappearance of *E(spl)mγ-GFP* only occurred in progeny born before the onset of imaging (*Figure 2B,D,F* and *Figure 2—video 2*), indicating that it must occur in cells that had been born from the NSC > 8 hr earlier. As the GMC divides after ~4 hr, it is likely that the cells regaining *E(spl)mγ-GFP* are progeny of a GMC division. Furthermore, in most cases they were intermediate in size between NBs and GMCs, suggesting that these changes precede the expression of *E(spl)mγ-GFP* (see P3.1 in *Figure 2B and D*). Subsequent to expressing *E(spl)mγ-GFP*, the cells adopted an asymmetric mode of division and only the larger daughter cell retained expression (see P2 in *Figure 2B and D*). These profiles indicate that NB-like cells arise from a reactivation of the stem-cell programme in the progeny, rather than an

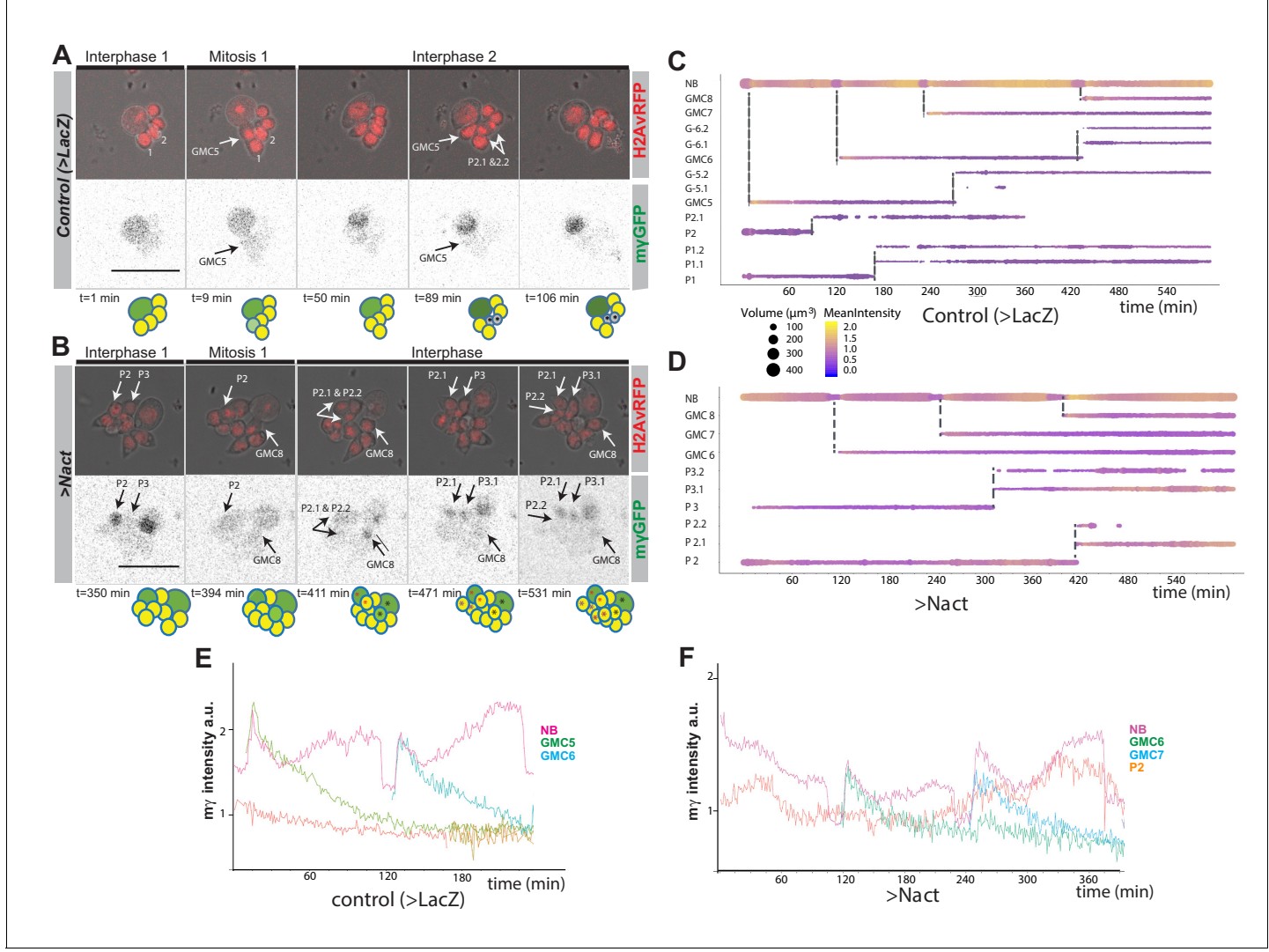

**Figure 2.** Live-imaging of E(spl)mγ-GFP expression in cultured NB lineages. (**A, B**) Time points from time lapse movie of (**A**) wild type (>LacZ) and (**B**) Nact expressing NB lineage, Histone-RFP (red, H2Av-RFP) marks all nuclei, corresponding E(spl)mγ-GFP levels in grey scale below. E(spl)mγ-GFP is rapidly extinguished in NB progeny in wild-type and in Nact expressing lineages but appears in some progeny after a delay. Time is indicated below each panels and cartoons summarise the cell populations in each lineage. NBs are show in darker shades of green, GMCs in yellow, newly born GMCs in pale green and neurons in grey. NB-like cells are shown in green. Coloured asterisks indicate cells coming from the same ancestor. White arrows indicate cells arising from a mitotic division. Numbers correspond to different cells indexed in the following diagrams. Scale bar 25 μm. (**C, D**) Bar chart diagram showing the progression over time of each cell in a wild-type NB lineage (**C**) or of selected NB progeny over time from one Nact expressing lineage. The diameter of the bar represents cell size and the colour represents expression levels of E(spl)mγ-GFP according to the scale (Blue, low to yellow, high). Dashed lines indicate mitotic events and the emerging daughter cells. (**E, F**) Diagrams depicting changes in the intensity of E(spl)mγ-GFP in NBs and their progeny over time from control (**E**) and Nact-expressing (**F**) NB lineages. Note the decay in E(spl)mγ-GFP levels in the newly born GMCs in both control and Nact lineages and the re-expression of E(spl)mγ-GFP in an older Nact progeny P2.

DOI: https://doi.org/10.7554/eLife.41637.006

The following videos are available for figure 2:

**Figure 2—video 1.** Time lapse movie of control (>*LacZ*) NB lineage with a duration of 10 hr.

DOI: https://doi.org/10.7554/eLife.41637.007

**Figure 2—video 2.** A 14 hr time-lapse movie of Nact expressing NB lineage.

DOI: https://doi.org/10.7554/eLife.41637.008

expansion of the stem cells themselves, and that the normal process of stem-cell fate attenuation is initiated but subsequently overcome.

## Mi-2 attenuates the response to Notch in NSC lineages

To identify the mechanisms that attenuate the response to Notch, delaying the transformation towards tumourigenesis, we screened for genes whose depletion by RNAi was sufficient to acceler-ate Notch-induced tumourigenesis. The candidates were primarily drawn from the pool of genes that are frequently mutated in human solid brain tumours (e.g neuroblastomas and glioblastomas) based on genome-wide sequencing studies (*Bosse and Maris, 2016*; *Chmielecki et al., 2017*; *Huether et al., 2014*; *Lin et al., 2016*; *Parsons et al., 2011*; *Pugh et al., 2013*; *Wu et al., 2014*). To probe function in the context of Notch-induced tumours, we used *GrhNB-Gal4* to drive Nact in com-bination with RNAi lines targeting candidate genes and monitored the consequences on Dpn. Expression was restricted to a 24 hr window, using Gal80ts, a condition where constitutively Nact produces only mild hyperplasia, as described above (*Figure 1A*).

From a total of 124 genes screened in this way, only a small proportion (13.7%) modified the Notch-induced hyperplasia. Of those, the most striking effect was seen with the knock down of *Mi-2*, a member of the NuRD ATP-remodelling complex. This significantly enhanced the number of Dpn-positive (Dpn$^{+ve}$) cells elicited by Nact: the total number of Dpn expressing cells was enhanced 2.7-fold when Mi-2 was depleted compared to Nact alone (1322 ± 71 vs 485 ± 27; *Figure 3A and D*). In contrast, Mi-2 depletion alone was not sufficient to bring about any increase in Dpn$^{+ve}$ cells (161 ± 2 vs 158 ± 2), despite that this treatment robustly extinguished Mi-2 protein levels in the NB lineages (*Figure 3—figure supplement 1A*). Similar effects, where Mi-2 depletion enhanced the hyperplasia from Nact, were obtained using a range of different Gal4 lines that drive expression in the NB and its early born progeny (*Figure 3—figure supplement 1B,C,E,F*). No such effects were seen when active Notch and *Mi-2 RNAi*, were co-expressed in more mature neuronal progeny (*Fig-ure 3—figure supplement 1D,G*), indicating there is a limited window during which they can revert these cells to NBs.

As Mi-2 is a core member of the NuRD chromatin remodelling complex, we next asked whether loss of other members of this complex would exhibit similar effects in the Notch driven hyperplasia. Knock down of *MTA-like* (*Figure 3B*) or *Caf1-p55* (*Figure 3C*) both produced a 1.4-fold enhance-ment of Notch-induced hyperplasia (*Figure 3E,F*). The fact that these phenotypes are relatively weak, compared to that from depleting Mi-2 may be technical, due to differences in knock-down or protein stability, or it may indicate that the other subunits, which are involved in protein-protein interactions, are less critical than the catalytic Mi-2. Nevertheless, the results suggest that Mi-2 func-tions in attenuating Notch activity via its participation in the NuRD complex.

To investigate whether the increase in Dpn$^{+ve}$ cells occurs because Mi-2/NuRD complex is neces-sary to switch off the response to Notch in GMCs, we performed live imaging of *E(spl)mγ-GFP* in NSC lineages with reduced Mi-2 with and without Nact. Depletion of Mi-2 alone had no discernible effect on *E(spl)mγ-GFP* in GMCs: it was extinguished as in normal conditions (*Figure 4—figure sup-plement 1* and *Figure 4—video 1*). However, in the context of Nact there was a striking persistence of *E(spl)mγ-GFP* in the GMCs in a substantial proportion of lineages (27.8%; *Figure 4A–E* and *Fig-ure 4—video 2,3*). Thus, the newly born GMCs retained high levels of *E(spl)mγ-GFP* for the full dura-tion of the movie, while the NB underwent several rounds of division (*Figure 4A–D* and *Figure 4—video 2,3*). Furthermore, when *E(spl)mγ-GFP* expressing progeny then divided, both daughter cells retained *E(spl)mγ-GFP* unlike the situation in the presence of Nact alone, where the acquisition of *E(spl)mγ-GFP* in the progeny was linked to them adopting asymmetrical divisions. These data demon-strate therefore that Mi-2/NuRD complex has an important role in suppressing the ability of the stem-cell progeny to respond to Notch and suggests that it contributes to the shut-down of stem-cell promoting genes.

## Mi-2 loss leads to de-repression of E(spl)-C genes

To investigate further whether Mi-2 is important for attenuating the response of Notch regulated enhancers, we analysed its effects at the *E(spl)*-Complex in Kc cells, where we have previously ana-lysed the changes in expression and in the recruitment of the Notch pathway transcription factor, Suppressor of Hairless [Su(H)], in the presence and absence of Notch activity (*Skalska et al., 2015*).

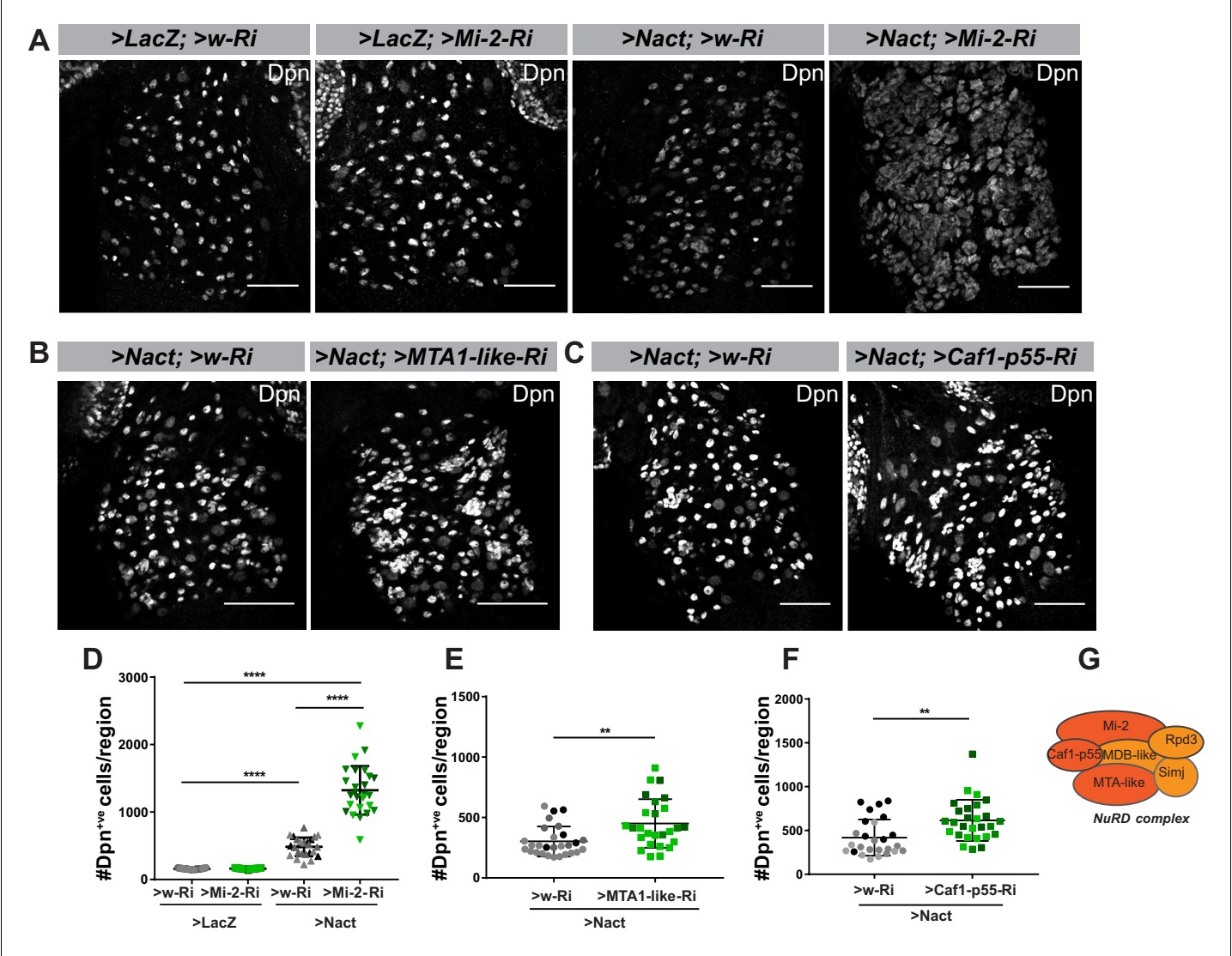

**Figure 3.** Mi-2 depletion exacerbates Notch-induced tumorigenesis. (**A**) Depletion of Mi-2 increases the number of hyperplastic Dpn[+ve] cells (white) caused by Nact expression in type I lineages. Expression of Dpn in wild type (>LacZ;>w Ri, first column), Mi-2 depleted only (>LacZ;>Mi-2-Ri; BL33415, 2nd column), Nact expressing (>NΔecd;>w Ri; 3rd column) and Nact with Mi-2 depleted (>NΔecd;>Mi-2-Ri, 4th column). (**B–C**) Loss of MTA-like (**B**) or Caf1-p55 (**C**) increases the number of hyperplastic Dpn[+ve] cells (white) caused by Nact expression. Expression of Dpn (white) in Nact expressing lineages (1st column: >NΔecd; >w-Ri) and in Nact with MTA-like or Caf1p55 depleted (2nd column: >NΔecd; >MTA-like-Ri; or >NΔecd; >Caf1 p55-Ri). (**D**) Number of Dpn[+ve] cells per VNC induced by expression of Nact was significantly increased by depletion of Mi-2, whereas depletion of Mi-2 alone had little effect. (**E–F**) Number of Dpn[+ve] cells per VNC induced by Nact was significantly increased upon knock down of MTA-like (**E**) or Caf1-p55 (**F**). Scatter dot plots where narrow black lines represent IQR, wider black line indicates median and whiskers indicate ±1.5 × IQR. (*p<0.05, **0.001 < P < 0.05, ***p<0.001, ****p<0.0001, t-test). N = 25–28 for each genotype tested; light and darker shades indicate data points from the three independent experiments. (**G**) Schematic depiction of NuRD Complex, with subunits tested here in dark orange.

DOI: https://doi.org/10.7554/eLife.41637.009

The following figure supplements are available for figure 3:

**Figure supplement 1.** Mi-2 depletion in NBs, GMCs or newly born progeny exacerbates Notch-induced tumorigenesis but Mi-2 depletion in older neurons does not.
DOI: https://doi.org/10.7554/eLife.41637.010

**Figure supplement 2.** Mi-2 depletion enhances the activation of Notch regulated genes.
DOI: https://doi.org/10.7554/eLife.41637.011

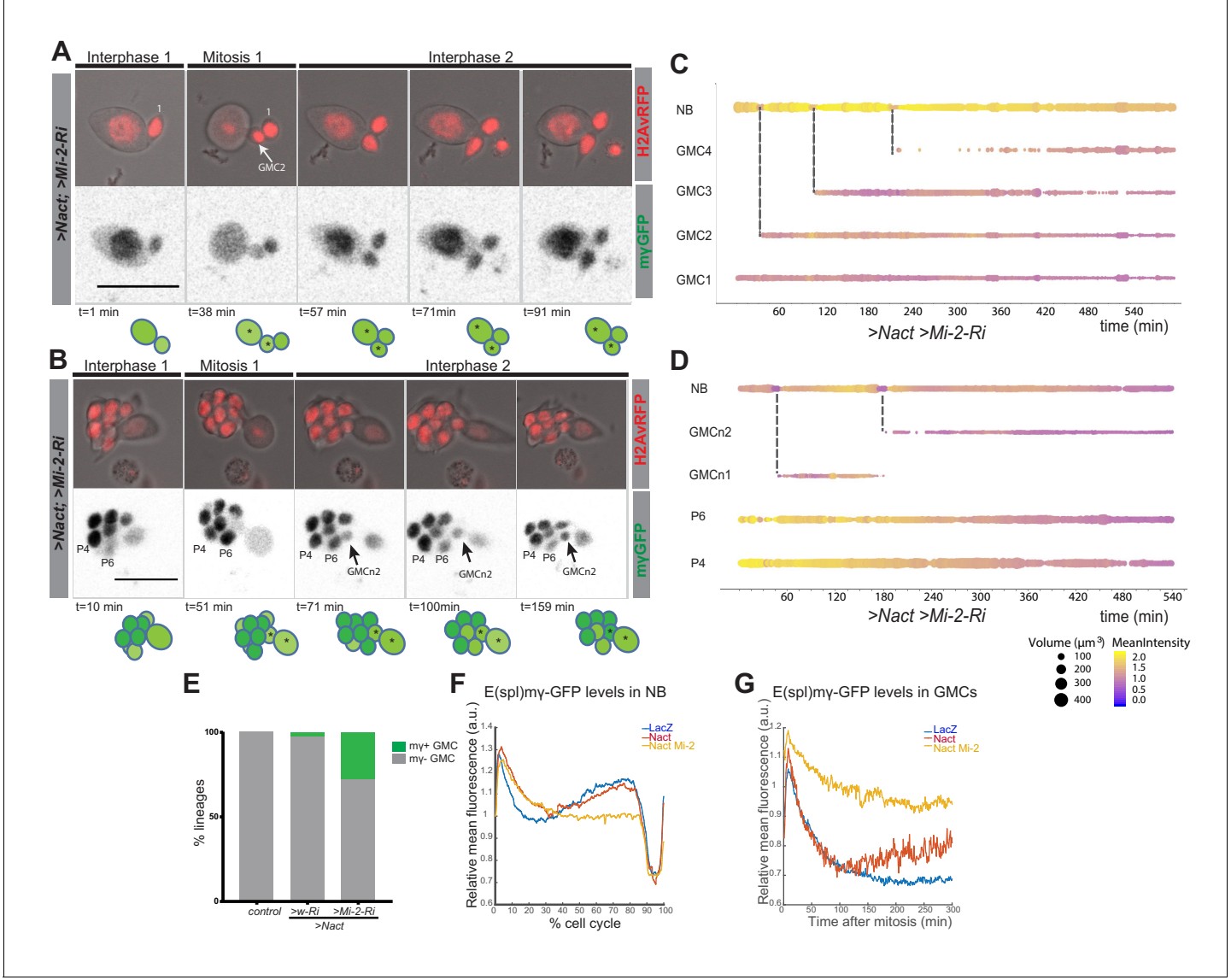

**Figure 4.** Expression of E(spl)mγ-GFP in GMCs following Mi-2 depletion in Nact expressing lineages. (**A, B**) Time points from two different time lapse movies of Nact Mi-2-RNAi NB lineages (dissociated from larval brains with *grhNBGal4 Gal80ts* at the permissive temperature for 24 hr). *E(spl)mγ-GFP* remains at high levels in emerging GMCs. Upper panels, bright-field images of NB and its progeny combined with Histone-RFP (red, H2Av-RFP) to monitor cell cycle stages; lower panels, expression of *E(spl)mγ-GFP* (greyscale). Time is depicted below each panel along with cartoons of the lineages where NBs are in darker shades of green, GMCs with low expression of *E(spl)mγ-GFP* in light green. Numbers correspond to different cells indexed in C, D. Scale bar 25 μm. (**C, D**) Bar chart depicting progression of each cell in a Nact *Mi-2 RNAi* NB lineage, bar thickness indicates cell-size and the colour represents *E(spl)mγ-GFP* levels according to the scale (blue low levels, yellow high levels). Dashed lines mark mitotic events linking to the emerging daughter cells. (**E**) Bar diagram depicted the percent of lineages with newly born GMCs that retain expression of *E(spl)mγ-GFP* in WT, Nact expressing and Nact expressing with compromised Mi-2. (**F**) Levels of *E(spl)mγ-GFP* during one division cycle in NBs from WT,>Nact and>Nact, *Mi-2 RNAi*. *E(spl)mγ-GFP* levels are high in interphase, immediately after mitosis in early G1 and before mitosis in late G2. However, in >Nact, Mi-2 knockdown, the second phase of high *E(spl)mγ-GFP* is lost. (**G**) Plot of *E(spl)mγ-GFP* levels in newly born GMCs in WT,>Nact and>Nact; *Mi-2 RNAi*. Note that *E(spl)mγ-GFP* levels decay in newly born GMCS in WT and >Nact but are maintained in >Nact; *Mi-2 RNAi* lineages.
DOI: https://doi.org/10.7554/eLife.41637.012

The following video and figure supplement are available for figure 4:

**Figure supplement 1.** Expression of E(spl)mγ-GFP in GMCs upon Mi-2 depletion in NB lineages.
DOI: https://doi.org/10.7554/eLife.41637.013

**Figure 4—video 1.** Time-lapse movie of *Mi-2-RNAi* NB lineage with a duration of 10 hr.
DOI: https://doi.org/10.7554/eLife.41637.014

**Figure 4—video 2.** Time lapse movie of Nact *Mi-2-RNAi* NB lineage with a duration of 10 hr.

*Figure 4 continued on next page*

*Figure 4 continued*

DOI: https://doi.org/10.7554/eLife.41637.015

**Figure 4—video 3.** Time lapse movie of a larger Nact *Mi-2-RNAi* NB lineage with a duration of 10 hr.

DOI: https://doi.org/10.7554/eLife.41637.016

Down-regulating Mi-2 by treating cells with RNAi resulted in a robust increase in the expression of *E (spl)mβ*, *E(spl)m3* (*Figure 5A–C*). These effects were similar to those seen when the co-repressor Hairless was depleted (*Figure 5C,D* and *Figure 5—figure supplement 1A*). Furthermore, loss of Mi-2 in the context of conditions where Notch signalling was activated, led to an enhanced induction of the *E(spl)* genes that are normally up-regulated in these conditions (*Figure 5C*). Similar effects were observed for three other Notch regulated genes in Kc cells, which were up-regulated following depletion of Mi-2 (*Figure 5—figure supplement 1B*). Thus, loss of Mi-2 leads to de-repression of *E (spl)* genes and of other Kc cell Notch targets in Notch off conditions and enhances their activation in Notch on conditions.

The effects of Mi-2 depletion are similar to those when Hairless is removed. Thus, one model is that Mi-2 is normally recruited by the Su(H)/Hairless corepressor complex to shut down target enhancers. If Hairless is required to recruit Mi-2, the combined knock down should produce similar effects to knockdown of Hairless alone. In contrast, concomitant knock down of both Mi-2 and Hairless led to a much greater de-repression of *E(spl)mβ* and *E(spl)m3* than from either knock down alone, suggesting the two factors work independently to repress these genes (*Figure 5E* and *Figure 5—figure supplement 1C*). In agreement, no interaction between Su(H) and Mi-2 was detected (*Figure 5F*) arguing that Mi-2 acts independently of the SuH-Hairless co-repressor complex to attenuate the expression of *E(spl)* genes. A second hypothesis was that Mi-2 reorganises the chromatin at target enhancers, preventing enhanced recruitment of Su(H) in the presence of Nact (*Kreher et al., 2017*). However, loss of Mi-2 had no effect on Su(H) recruitment in either Notch off or Notch on conditions (*Figure 5G*) neither did it affect the levels of H3K27 trimethylation, the repressive histone modification deposited by Polycomb complexes, or of H3K27ac (*Figure 5—figure supplement 1D, E*).

These data raise the question whether the Mi-2 complex acts directly at the target enhancers or exerts its effects indirectly. We therefore examined the overlap between Notch regulated enhancers and Mi-2-bound regions based on previous genome-wide profiling of Su(H) (*Skalska et al., 2015*) and Mi-2 (*Ho et al., 2014*) by chromatin immunoprecipitation in Kc cells. Strikingly, 72% of Su(H) bound regions overlapped with Mi-2 occupied sites. These included significant enrichments for Mi-2 at the *E(spl)mβ* and *E(spl)m3* regulatory regions in the *E(spl)-C* (*Figure 5A*), which we verified by ChIP qPCR (*Figure 5H*). Thus, Mi-2 is present at Notch regulated enhancers suggesting it has a direct effect on their activity.

In summary, Mi-2 binds directly to Notch target genes and exerts a repressive effect on them, even in Kc cells, but it appears to do so independently of Su(H) and Hairless. This implies that other factors must be involved in its recruitment.

## Zfh1 cooperates with Mi-2 in stem-cell lineages

If Mi-2 recruitment is not dependent on Su(H), this implies that other DNA-binding transcription factors are required. Likely candidates in the NB lineages include transcription factors that are required to promote differentiation or that have been linked to Mi-2. The former include Prospero, which is asymmetrically segregated and accumulates in the nuclei of the GMCs, and Lola, which is highly expressed in stem-cell progeny and is necessary to prevent their de-differentiation (*Southall et al., 2014*). The latter include Tramtrack, a BTB zinc-finger protein shown to bind directly to Mi-2 (*Murawsky et al., 2001*; *Reddy et al., 2010*), and Zfh1, a zinc-finger- homeodomain repressor whose homologue ZEB2 cooperates with NuRD complex to antagonise Notch in Schwann cells (*Wu et al., 2016*). We therefore tested whether depletion of any of the candidate factors was sufficient to enhance Nact-induced hyperplasia in a similar way to Mi-2. Of those tested, only two factors significantly enhanced the hyperplasia, namely Prospero and Zfh1, indicating that they were plausible candidates to recruit Mi-2 (*Figure 6A,B* and *Figure 6—figure supplement 1A,B*).

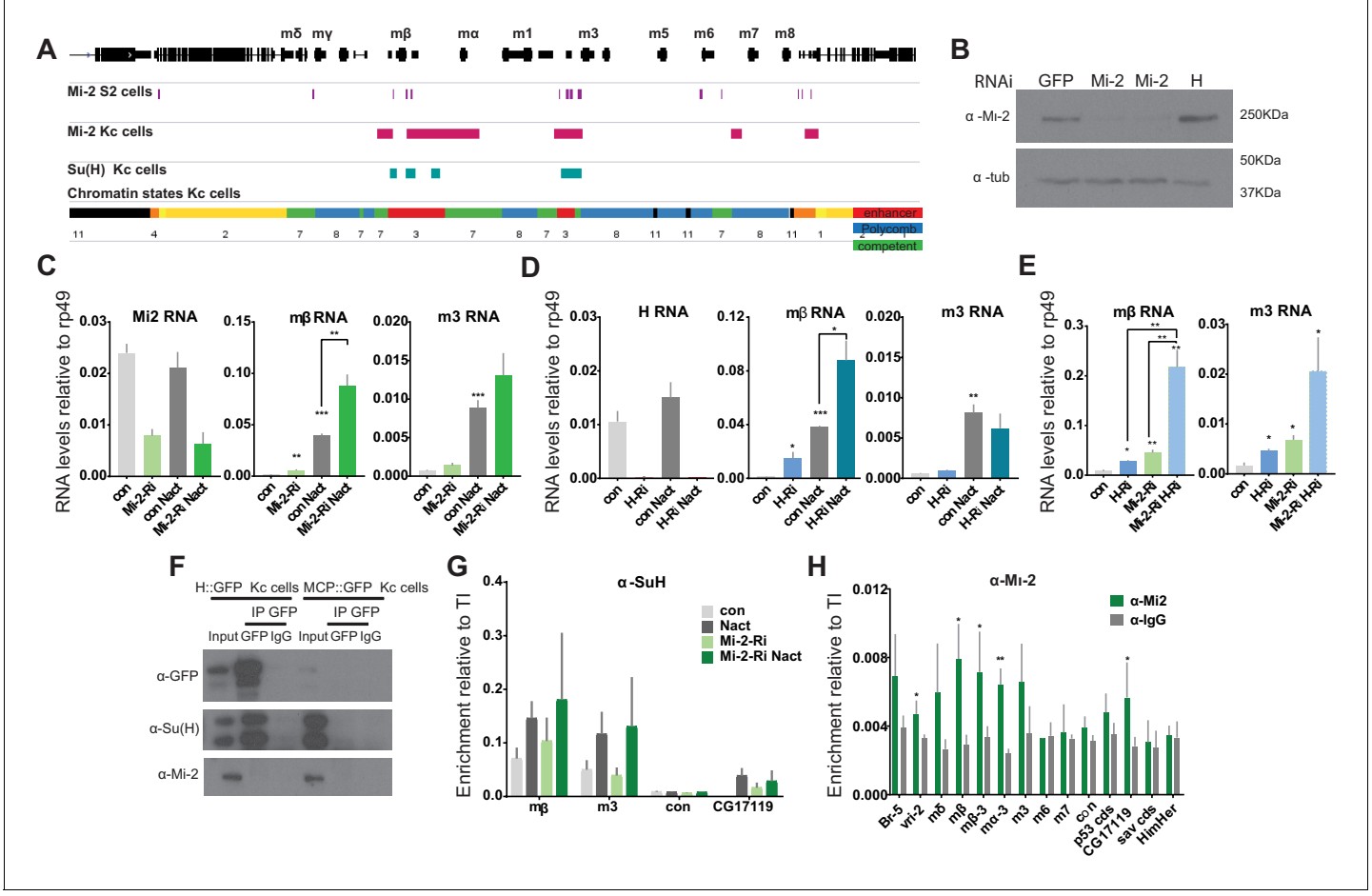

**Figure 5.** Loss of Mi-2 leads to de-repression of *E(spl)-C* genes. (A) Genomic region spanning *E(spl)-C* locus with graphs depicting Mi-2 bound regions in S2 cells (purple) (*Kreher et al., 2017*) and Kc cells (magenta) (modEncode; *Ho et al., 2014*), Su(H)-bound regions in Kc cells (cyan) (*Skalska et al., 2015*) and chromatin signatures from Kc cells (*Skalska et al., 2015*). Gene models are depicted in black. (B) Levels of Mi-2 protein are reduced upon knockdown of Mi-2 via RNAi in Kc cells for 3 days compared to knockdown of GFP or *Hairless* via RNAi. Anti-tubulin is a control for loading. (C, D) Fold change in RNA levels in Kc cells upon knockdown of *Mi-2* (C) or *Hairless* (D) compared to control conditions (con: GFP RNAi) and to cells with Notch activation (Nact). Note that *E(spl)mβ* and *E(spl)m3* are both significantly de-repressed in Notch-off state and show even higher increase in expression in Notch-on state. (E) Fold change in RNA levels in Kc cells upon combined knockdown of Mi-2 and Hairless compared to control conditions and single knockdown of Mi-2 or Hairless. Note additive effects on *E(spl)mβ* and *E(spl)m3* de-repression in combined knock-down conditions. (F) Mi-2 does not directly interact with Hairless. Immunoprecipitations with anti-GFP from Kc cells expressing Hairless-GFP or MCP-GFP as control, Su(H) is co-purified with Hairless but not Mi-2. (G) Enrichment of Su(H) is indicated at *E(spl)-C* or regions in Kc cells as revealed by ChIP in control (grey) or Mi-2 knockdown (for 3 days, green) conditions in Notch off (light shading) and Notch active (EGTA treatment 30 min; dark shading). Su(H) recruitment in Notch active and in control conditions is not altered by knockdown of Mi-2. (H) Enrichment of Mi-2 at indicated positions in *E(spl)-C* or other regions in Kc cells as revealed by ChIP. (P values: *0.01 < P < 0.05, ** 0.001 < P < 0.01, ***p<0,001, Multiple t-tests).

DOI: https://doi.org/10.7554/eLife.41637.017

The following figure supplement is available for figure 5:

**Figure supplement 1.** Effects of Mi-2 depletion on histone modifications at the *E(spl)-C* locus.
DOI: https://doi.org/10.7554/eLife.41637.018

If Prospero or Zfh1 are involved in attenuating the activity of stem-cell genes via their recruitment of Mi-2, their ectopic expression in NBs might promote their differentiation in a Mi-2-dependent manner. Indeed, expression of either Prospero or Zfh1 was sufficient to significantly reduce the number of Dpn[+ve] cells, suggesting they have suppressed NB identity (*Figure 6C,D* and *Figure 6—figure supplement 1C,D*). Furthermore, depletion of Mi-2 partially overcame the effects of Zfh1 overexpression, restoring the number of NBs to close to normal levels (*Figure 6B and D*). In contrast, Mi-2 depletion failed to rescue the effects from Prospero over-expression (*Figure 6—figure*

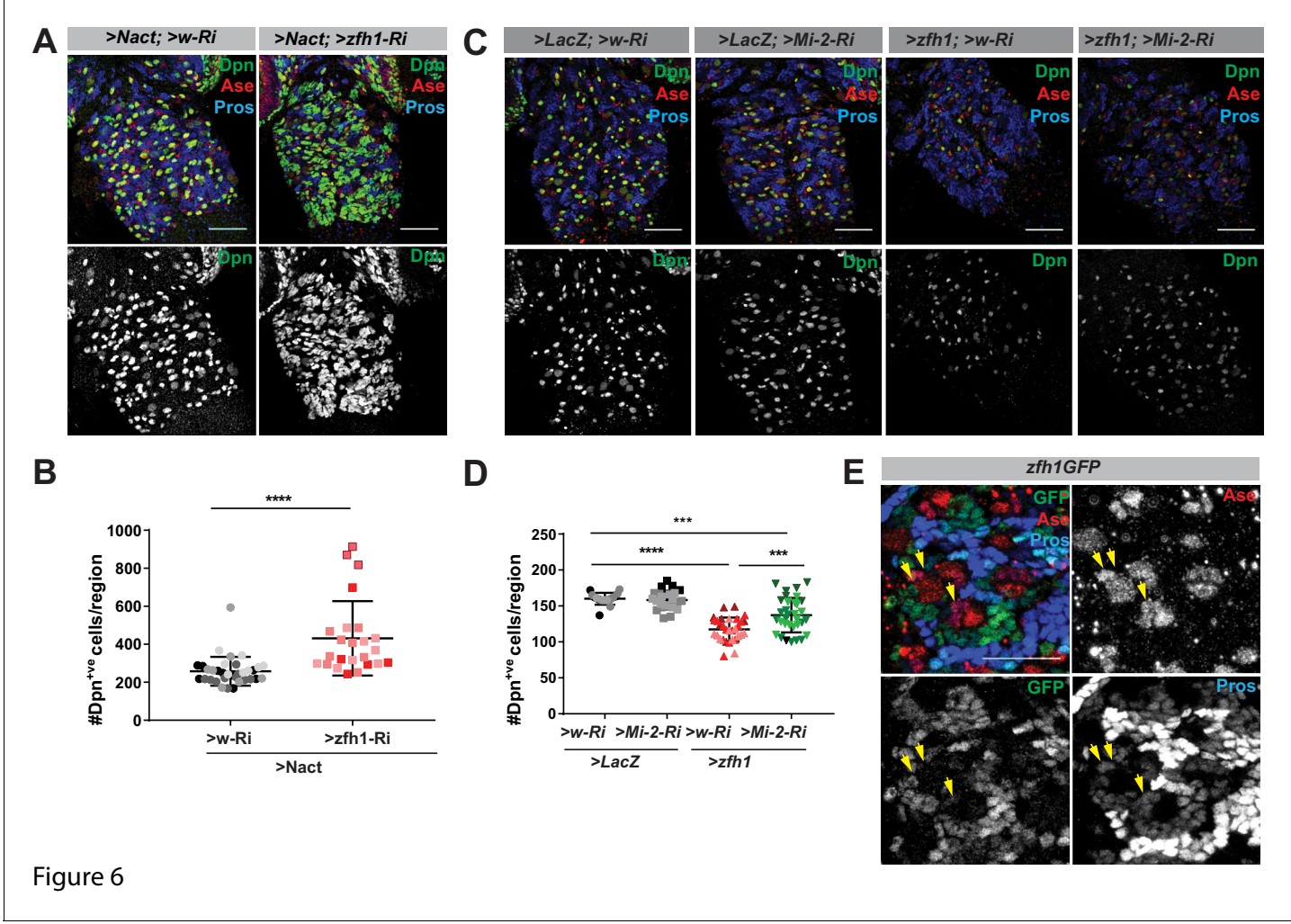

Figure 6

**Figure 6.** Cooperation between Zfh1 and Mi-2 in NB lineages. (**A**) Zfh1 depletion enhances the number of Dpn[+ve] cells (green or white channels) caused by Nact over-expression, Ase marks NBs and GMCs, Pros marks progeny. (**B**) Scatter dot plot, the number of Dpn[+ve] cells per VNC in >Nact; >zfh1 Ri (N = 24, three experiments) was significantly increased compared to >Nact;>wRi larvae (N = 34, three experiments). (**C**) Overexpression of Zfh (>zfh1 PB; w–Ri) results in decreased number of Dpn[+ve] cells per VNC (N = 31, three experiments) compared to wild type (>LacZ;>w Ri; N = 17, three experiments) or Mi-2 knockdown alone (>LacZ;>Mi-2-Ri; N = 25, three experiments). Concomitant loss of Mi-2 (>zfh1 PB;>Mi-2Ri; N = 33, three experiments) rescues Zfh1-induced loss of NBs. Ase and Pros as in (**A**). (**D**) Scatter dot plot with number of Dpn[+ve] cells in each condition quantified. (**E**) *zfh1-GFP* expression in type I NB lineages. Zfh1 is expressed in newly born neurons and at low levels in GMCs (yellow arrowheads). Ase marks neuroblasts and GMCs whereas Pros marks older neuronal progeny in NB lineages.

DOI: https://doi.org/10.7554/eLife.41637.019

The following figure supplements are available for figure 6:

**Figure supplement 1.** Prospero does not cooperate with Mi-2 in NB lineages.
DOI: https://doi.org/10.7554/eLife.41637.020
**Figure supplement 2.** Zfh1 resists Notch-induced tumorigenesis.
DOI: https://doi.org/10.7554/eLife.41637.021

*supplement 1C,D*). Finally, if Zfh1 antagonises the ability of Notch to promote expression of stem-cell genes, its ectopic expression in NBs with excessive Notch should prevent the formation of hyperplasia. In agreement, Zfh1 overexpression attenuated the Notch-induced hyperplasia, an effect that was partially abolished by loss of Mi-2 (*Figure 6—figure supplement 2A–B*) Together these data argue that Zfh1 cooperates with Mi-2 to suppress stem-cell identity.

If Zfh1 is involved in shutting down the stem-cell programme with Mi-2, it should be present in the NSC progeny during the period where they are resistant to ectopic Notch activity. Indeed, using

a fly line in which the endogenous Zfh1 is fused to GFP (*Albert et al., 2018*). Zfh1 was detected in the progeny that are located in proximity to the NB and in the Asense-labelled GMCs, albeit at lower levels (*Figure 6E*) in Type I NB lineages. In contrast, Zfh1 was absent from the NB and INPs in Type II lineages albeit these cells exhibited strong or moderate Mi-2 levels, respectively (*Figure 6— figure supplement 2C*). Together the results point to Zfh1 as an important intermediary in recruiting Mi-2/NuRD complex to shut off stem-cell enhancers in the NB progeny of Type I lineages. However, it is likely that different factors will recruit Mi2 in other contexts where it is required to tune Notch-responsive enhancers.

## Discussion

In order to progress towards differentiation, stem-cell progeny need to shut off the transcription of genes involved in maintaining the self-renewing stem-cell programme. Our results demonstrate the importance of Mi-2/NuRD chromatin complex in making this switch in *Drosophila* NB lineages, specifically in Type I lineages. Under normal conditions, these stem-cell progeny are remarkably resistant to ectopic Notch activity, only reverting to a more-stem-cell-like fate after prolonged exposure to a constitutively active Notch. However, when Mi-2 or other members of the NuRD complex were depleted, progeny cells reverted with much higher frequency when exposed to high levels of Notch activity, leading to rapid onset of hyperplasia. Thus, we propose that Mi-2/NuRD complex functions to decommission the stem-cell enhancers in the differentiating progeny (*Figure 7*) rendering them resistant to stem-cell promoting factors like Notch (*Whyte et al., 2012*). As the iterative use of signaling pathways, including Notch, is important throughout development to generate different tissue and cell types, it is likely that NuRD will be widely deployed in such developmental transitions to decommission enhancers in readiness for subsequent lineage decisions.

The fact that expression of Nact fails to sustain expression of even its direct target genes, *E(spl) mγ-GFP* and *dpn*, in the progeny GMCs argues that their enhancers have become refractory. As the multi-protein NuRD complex contains both the Mi-2/CHD4 ATP-dependent chromatin remodelling and histone deacetylase subunits it could curtail enhancer activity by promoting an increase in local nucleosome density at these enhancers or by depleting H3K27 actetylation, as occurs in mouse embryonic stem cells where Mi-2 enables recruitment of Polycomb repression complexes (*Reynolds et al., 2012b*). Based on the results from analysing chromatin changes following Mi-2 knock-down in cultured cells, we favour the former model. No change in H3K27ac or H3K27me3 was observed at Notch target enhancers despite the fact that the genes were de-repressed. Furthermore, there was no evidence for eviction of the key transcription factor Su(H) from these enhancers. This is more compatible with a model where NuRD remodelling perturbs the recruitment of Mediator and PolII to restrict initiation from these genes (*Bornelöv et al., 2018*).

Although originally considered a repressive complex, NuRD is found at most sites of active transcription in embryonic stem cells, where it appears to fine-tune gene expression as well as having a major role in allowing cells to exit self-renewal (*Bornelöv et al., 2018*; *de Dieuleveult et al., 2016*; *Miller et al., 2016*; *Reynolds et al., 2012a*; *Reynolds et al., 2012b*). It remains unclear what triggers the changes that enable this transition towards differentiation. One possibility is that the first step towards decommissioning or repurposing these enhancers involves removal of positive factors that keep a 'check' on NuRD activity with the consequence that NuRD predominates to increase local nucleosome density (*Whyte et al., 2012*). At the same time, the upregulation of differentiation factors that favour NuRD complex recruitment can enhance such a switch. This is the mechanism we propose for Zfh1, which is upregulated in the NB progeny and whose ability to suppress the stem-cell programme when ectopically expressed appears to require Mi-2 (*Figure 7*). Depletion of Zfh1 in the context of ectopic Nact also enhanced hyperplasia, similarly to depletion of Mi-2. Although there are caveats to using RNAi for establishing epistatic relationships, the data argue that Zfh1 and NuRD act together to render enhancers of genes such as *E(spl)mγ-GFP* and *dpn* refractory to Notch activity. This is similar to the role proposed for the mammalian homologue ZEB2 during Schwann Cell differentiation, where it engages the NuRD complex to antagonise Notch and Sox2 (*Wu et al., 2016*). Indeed, a missense mutation in the Zeb2 that abolished its association with the NuRD complex rendered it incapable of promoting Schwann Cell differentiation (*Wu et al., 2016*).

Other factors have also been shown to block responsiveness to Notch. For example eyeless/Pax6 is switched on in the progeny of old Type II NBs and makes them refractory to Notch

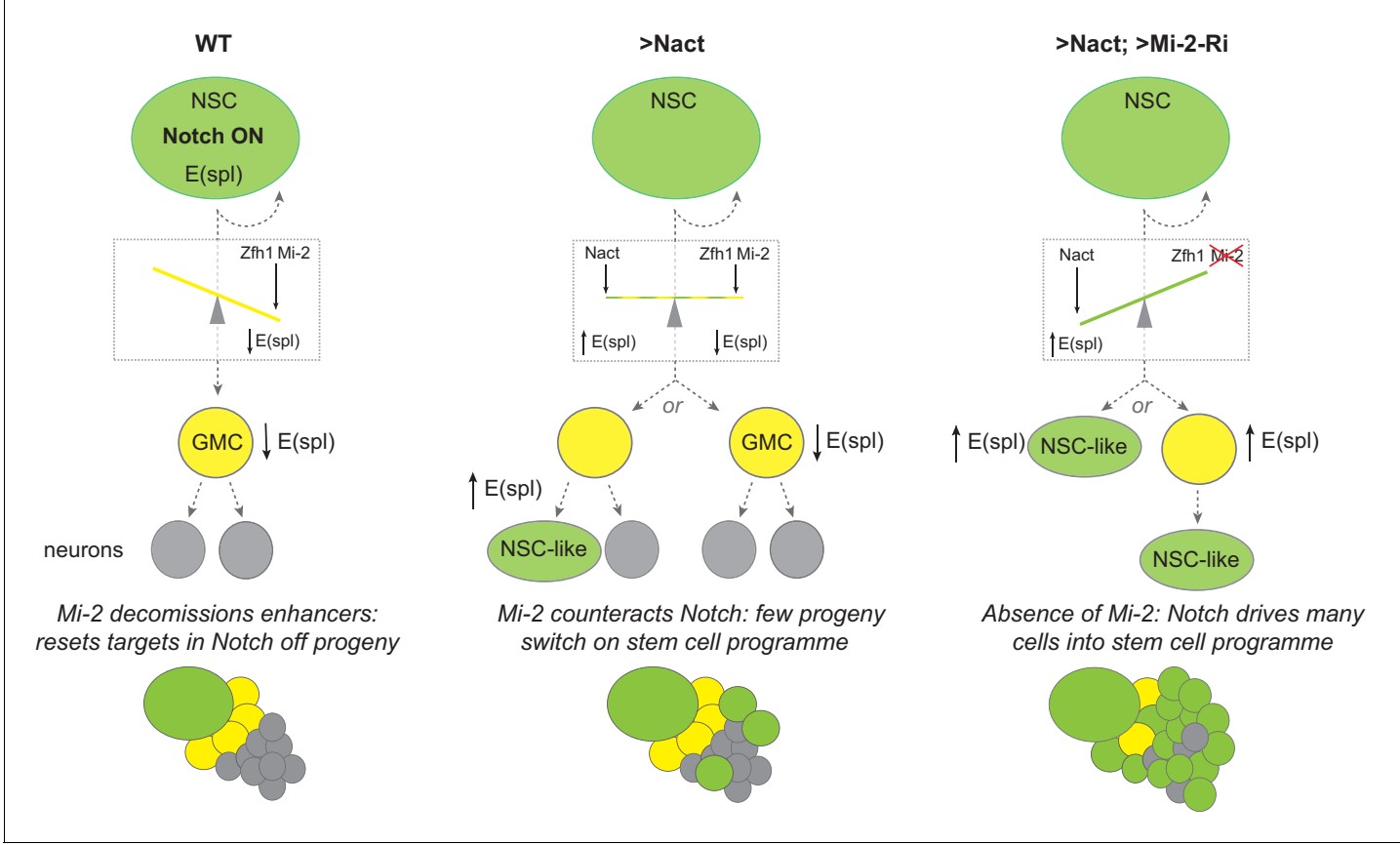

**Figure 7.** Model summarizing the role of Mi-2 in decommissioning Notch-responsive enhancers in NB progeny. The presence of Zfh1 and Mi-2 favours the decommissioning of enhancers from E(spl) and other Notch target genes in GMCs (yellow) to ensure their expression is switched off. Notch is on in NBs (green) and off in GMCs (yellow cells) due to asymmetrical segregation of Numb (ref). GMCs divide to produce two post-mitotic neuronal cells (grey). In NB lineages with constitutive Notch activity (Nact), the presence of Mi-2 at enhancers, recruited by Zfh1 (and potentially by other factors too), is sufficient to attenuate Nact, so that E(spl) and other target genes are switched off in GMCs. In a few NB progeny the effects of Mi-2 are overcome with time, and E(spl) genes are up-regulated as they revert to NB-like cells. Depletion of Mi-2 in NB lineages expressing Nact severely compromises enhancer decommissioning so that E(spl) and other Notch target genes are upregulated in many of the GMCs. The majority of the NB progeny acquire an NB-like fate. SUPPLEMENTARY MATERIAL: Legends for videos.
DOI: https://doi.org/10.7554/eLife.41637.024

(*Farnsworth et al., 2015*) and Ikaros expression during T-cell differentiation shapes the timing and repertoire of Notch target genes (*Zhang et al., 2011*). Whether these factors rely on Mi-2 remains to be explored but as Ikaros was found to associate with the Mi-2/NuRD complex, it could rely on similar mechanism to Zfh1/Zeb2 (*Sridharan and Smale, 2007*). Likewise, PROX 1, the mammalian homologue of Prospero, interacted with NuRD complex to suppress Notch pathway in colorectal cancer cells. We propose therefore that different factors will be involved in recruiting Mi-2 depending on the cell-type. For example, in Kc cells the Su(H) bound regions are enriched for the binding motif for Tramtrack, which has been shown to physically interact with Mi-2 (*Murawsky et al., 2001*) and is thus a good candidate. Conversely, the fact that Type II neuroblast lineages are exquisitely sensitive to ectopic Notch activity argues that these cells lack factor(s) that recruit Mi-2 to the target enhancers. Indeed Zfh1 is not expressed in the intermediate progeny of these lineages and it has been shown that an alternate mechanism, involving the repressor earmuff/dFezf is deployed to repress the competence of INPs to respond to Notch targets and thus their dedifferentiation into NBs (*Janssens et al., 2014*; *Koe et al., 2014*). .

The fact that prolonged Notch activity can ultimately reprogramme progeny towards the stem-cell fates, driving hyperplasia even in the presence of Mi2, argues that NuRD is not fully silencing target genes. This fits with the model that NuRD is involved in fine-tuning transcriptional responses

(*Bornelöv et al., 2018*). Thus, at enhancers where NuRD predominates it would increase nucleosome density rendering the enhancer less accessible so that higher amounts of signal are required to overcome it. If active Notch accumulates in the older progeny, it could reach a critical threshold to win-out over NuRD. Alternatively, Mi2/NuRD may only be required during the acute phase of transitions and be replaced at later times with other modes of repression that can be more easily overcome by Notch activity. The fact that Zfh1 expression is not sustained in older progeny fits with this model. However, distinguishing these possibilities will require an understanding of the real-time dynamics of the chromatin changes at target loci in individual progeny.

# Materials and methods

## Key resources table

| Reagent type (species) or resource | Designation | Source or reference | Identifiers | Additional information |
|---|---|---|---|---|
| Gene (*D. melanogaster*) | Notch | NA | ID_FLYBASE: FBgn0004647 | |
| Gene (*D. melanogaster*) | Mi-2 | NA | ID_FLYBASE: FBgn0262519 | |
| Gene (*D. melanogaster*) | E(spl)mγ | NA | ID_FLYBASE: FBgn0002735 | |
| Gene (*D. melanogaster*) | dpn | NA | ID_FLYBASE: FBgn0010109 | |
| Gene (*D. melanogaster*) | zfh1 | NA | ID_FLYBASE: FBgn0004606 | |
| Genetic reagent (*D. melanogaster*) | grh-Gal4 | PMID: 9651493 | | *Prokop et al., 1998* |
| Genetic reagent (*D. melanogaster*) | UAS-NΔecd | PMID: 8413612, 8343959 | | *Fortini et al., 1993*; *Rebay et al., 1993* |
| Genetic reagent (*D. melanogaster*) | UAS-Mi-2-RNAi | Bloomington Stock Center | ID_BDSC:33419 | Genotype: y[1] sc[*] v[1]; P{y[+t7.7] v[+t1.8]=TRiP.HMS00301}attP2 |
| Genetic reagent (*D. melanogaster*) | UAS-MTA1-like RNAi | Bloomington Stock Center | ID_BDSC:33745 | Genotype: y[1] sc[*] v[1]; P{y[+t7.7]v[+t1.8]=TRiP.HMS01084}attP2 |
| Genetic reagent (*D. melanogaster*) | UAS-Caf1p-55 RNAi | Bloomington Stock Center | ID_BDSC:34069 | Genotype: y[1] sc[*] v[1]; P{y[+t7.7] v[+t1.8]=TRiP.HMS00051}attP2 |
| Genetic reagent (*D. melanogaster*) | UAS-zfh1-RNAi | Bloomington Stock Center | ID_BDSC:29347 | Genotype: y[1] v[1]; P{y[+t7.7] v[+t1.8]=TRiP.JF02509}attP2/TM3, Sb[1] |
| Genetic reagent (*D. melanogaster*) | UAS-Pros RNAi | Bloomington Stock Center | ID_BDSC: 26745 | Genotype: y[1] v[1]; P{y[+t7.7] v[+t1.8]=TRiP.JF02308}attP2/TM3, Sb[1] |
| Genetic reagent (*D. melanogaster*) | UAS-zfh1-RB | Bloomington Stock Center | ID_BDSC: 6879 | made by Antonio Postigo (*Siles et al., 2013*); genotype: w[1118]; P{w[+mC]=UAS-zfh1.P}2B |

*Continued on next page*

*Continued*

| Reagent type (species) or resource | Designation | Source or reference | Identifiers | Additional information |
|---|---|---|---|---|
| Genetic reagent (*D. melanogaster*) | H2Av-RFP | Bloomington Stock Center | ID_BDSC:23651 | w*; P {His2Av-mRFP1}II.2 |
| Genetic reagent (*D. melanogaster*) | zfh1-GFP | PMID: 30002131 | | *Albert et al., 2018* |
| Cell line (*D. melanogaster*) | Kc 167 cells | Drosophila Genomics Resource Center | ID_DGRC: 1 | |
| Antibody | guinea pig polyclonal anti-Deadpan | Christos Delidakis, Heraklion, Greece | | (Immuno fluorescence dilution 1:2000) |
| Antibody | rabbit polyclonal anti-Mi-2 | Alexander Brehm, Marburg, Germany | | (Immuno fluorescence dilution 1:10,000) *Kreher et al., 2017*; PMID: 28378812 |
| Antibody | rabbit polyclonal anti-Asense | Y.N.Yan, San Fransisco, USA | | (Immunofluorescence dilution 1:2000) *Brand et al., 1993*; PMID 8565817 |
| Antibody | mouse monoclonal anti-Pros | Drosophila Hybridoma Studies Bank | ID_DHSB: MR1A | (Immuno fluorescence dilution 1:50) |
| Antibody | mouse monoclonal anti-Mira | Fumio Matsuzaki, Kobe, Japan | | (Immuno fluorescence dilution 1:100) *Ohshiro et al., 2000*; PMID 11117747 |
| Antibody | goat polyclonal anti-Su(H) | Santa Cruz Biotechnology | ID_SC: sc15813 | (12 µl per ChIP with $15 \times 10^6$ cells) |
| Antibody | rabbit polyclonal anti-H3K56ac | Active Motif | ID_Active Motif: 39281 | (3 µl per ChIP with $15 \times 10^6$ cells) |
| Antibody | rabbit polyclonal anti-H3K27me3 | Millipore | ID_Millipore: 07–449 | (1 µl per ChIP with $15 \times 10^6$ cells) |
| Antibody | rabbit polyclonal anti-H3K27ac | Abcam | ID_ABCAM: ab4729 | (10 µl per ChIP with $15 \times 10^6$ cells) |
| Antibody | rabbit polyclonal anti-H3 | Abcam | ID_ABCAM: ab1791 | (1 µl per ChIP with $15 \times 10^6$ cells) |
| Sequence-based reagent | | This paper | | oligonucleotides for mRNA levels and ChIP qPCR assays; *Tables 1* and *2* |
| Commercial assay or kit | MEGAscript T7 Transcription Kit | ThermoFischer Scientific | ID_TFS: AMB 13345 | |
| Commercial assay or kit | Ambion,DNA-free kit | ThermoFischer Scientific | ID_TFS: AM1906 | |
| Commercial assay or kit | M-MLV reverse transcriptase | Promega Corporation | ID_Promega: M531A | |
| Commercial assay or kit | LightCycler 480 SYBR Green I Master PCR Kit | Roche | ID_Roche: 4707516001 | |
| Chemical compound, drug | Collagenase | Sigma | ID_Sigma: C0130 | |

## Drosophila genetics

*Drosophila* stocks are described in FlyBase and were obtained from the Bloomington, Vienna or Kyoto Stocks Centres unless otherwise indicated. Over-proliferating third instar larval CNSs were generated by crossing *UAS-NΔecd; UAS-wRNAi* flies with *tub-Gal80ts; grh-Gal4* flies to drive

**Table 1.** PCR primers for RNA analysis:

| Primer name (for RNA) | Primer sequence |
| --- | --- |
| mβ coding sequence for | AGAAGTGAGCAGCAGCCATC |
| mβ coding sequence rev | GCTGGACTTGAAACCGCACC |
| m3 coding sequence for | CGTCTGCAGCTCAATTAGTC |
| m3 coding sequence rev | AGCCCACCCACCTCAACCAG |
| mδ coding sequence for | AGGATCTCATCGTGGACACC |
| mδ coding sequence rev | CAGACTTCTTCGCCATGATG |
| mα coding sequence for | TCCCAATGCTCGCCTTTAGA |
| mα coding sequence rev | TGATCTCCAAGCGGAGTATG |
| mγ coding sequence for | TCAGATCCAGCCAGCAGAAA |
| mγ coding sequence rev | CTGGAGATTGGCGAAATGGG |
| m7 coding sequence for | GCACTGCACACACACACTTC |
| m7 coding sequence rev | AACAATATACGTGGCCGGTT |
| Rpl32 sense | ATGCTAAGCTGTCGCACAAATG |
| Rpl32 antisense | GTTCGATCCGTAACCGATGT |
| Mi-2 coding sequence for | GAGCGGCCTACCTTAACCTC |
| Mi-2 coding sequence rev | TCAGATGCTGATGGGATTCA |
| Hairless coding sequence for | TACGAGCGAGGATGAGGAAC |
| Hairless coding sequence rev | TCCCAATGCTCGCCTTTAGA |
| CG17119 coding sequence for | TCGTTGAGCATCACAGGATTCA |
| CG17119 coding sequence rev | TCAACTGCGGCCTCTATTTCAT |
| CG12290 coding sequence for | AACTGATGCCCGTACAGGAG |
| CG12290 coding sequence rev | GCTGTCTGGCGGAGTAGTTC |
| Notch coding sequence for | CGGACTCGACTGTGAGAACA |
| Notch coding sequence rev | GGAACTGAGCCTGAATCTCG |
| Rgl coding sequence for | GAGGATTGGCACGAGGATAA |
| Rgl coding sequence rev | ACTGTTTGATGAGCCGTTCC |

DOI: https://doi.org/10.7554/eLife.41637.022

expression in most NSCs (*Prokop et al., 1998*). Crosses were kept at 18°C for 7 days, then shifted to 30°C for 8 hr, 24 hr or 48 hr prior to dissection.

Flies for the live imaging experiments were of the following genotypes:

Control: *UAS-LacZ, E(spl)mγ-GFP; H2Av-RFP*,

Overexpressing active Notch: *UAS-NΔecd, E(spl)mγ-GFP; H2Av-RFP*

Depletion of Mi-2 in presence of Nact: *UAS-NΔecd, E(spl)mγ-GFP; UAS-Mi-2RNAi*

All above flies were crossed to *tub-Gal80ts, H2Av-RFP; grhNB-Gal4* flies. These crosses were kept at 18°C for 7 days, then shifted to 30°C for 24 hr prior to dissection and brain dissociation.

For knock down of members of the NuRD complex or differentiation TFs in NSCs, the following RNAi lines were initially combined with *UAS-NΔecd* or *UAS-LacZ* and then crossed to *tub-Gal80ts; grhNB-Gal4* flies: *UAS-w RNAi* (BL35573), *UAS-Mi-2 RNAi* (BL33419), *UAS-MTA1-like RNAi* (BL33745), *UAS-Caf1p-55 RNAi*(BL34069), *UAS-zfh1 RNAi* (BL29347), UAS-*Pros RNAi* (BL26745). Crosses were kept at 18°C for 7 days, then shifted to 30°C for 24 hr prior to dissection. Transgenic Flies carrying the putative Notch-regulated enhancers (NREs) in *Cyclin E* and *pathetic* (*Djiane et al., 2013*) were also crossed with the above genotypes to assess the effects of Mi-2 knockdown in other Notch target genes in Nact NB lineages.

For overexpression of TFs in NBs, the following lines: *UAS-lacZ* (control), or*UAS-zfh-RB* (made by Antonio Postigo (*Siles et al., 2013*); BL6879) were combined with *UAS-Mi-2 RNAi* or *UAS-w RNAi* (control) and crossed to *insc-Gal4; tub-Gal80ts. UAS-ProsYFP* (*Choksi et al., 2006*) or *UAS-LacZ*

**Table 2.** Primers for ChIP – qPCR:

| Primer name (for ChIP) | Primer sequence |
| --- | --- |
| mβ for | AGAGGTCTGTGCGACTTGG |
| mβ rev | GGATGGAAGGCATGTGCT |
| m3 for | ACACACACAAACACCCATCC |
| m3 rev | CGAGGCAGTAGCCTATGTGA |
| con for | CAATTCCACGAAGCACAGTC |
| conrev | GAGGAGCAGTCCATCGAGTT |
| CG17119_qPCR_5 | TACATGGGCTTTGTCGGTCG |
| CG17119_qPCR_3 | CACGGCCCTCGCCATATAAA |
| Br-5 for | CACAGAAGGAAGAAGCAGCA |
| Br-5 rev | CGGGACTGGCAAATTTCTT |
| Vri-2 for | TGTGGACGTGGAATTGGAT |
| Vri-2 rev | CAATGACACTTGGGCATGG |
| mδ for | AGCAGAAACCCACACCCATA |
| mδ rev | TTCCCTCGAGAAAAGAGAGC |
| mδ−3 for | AGACCAGAGACCCAGAGCAA |
| mδ−3 rev | GGCGCAATAAAGTTGAAAGC |
| mα−3 for | AAGCCAGTGGACTCTGCTCT |
| mα−3 rev | TGATCTCCAAGCGGAGTATG |
| m6 for | CGAACGTTGGGCTGATAGTT |
| m6 rev | AAAAGTCCAACCACCCAACA |
| m7 for | CAAGCATGCGCACACATATT |
| m7 rev | CATCGGGGTTGGCTTATTGT |
| Sav-cds-5 | GAGTAGGTGTTCCGACTGGTG |
| Sav-cds-3 | ATCAGCGGGCCAAGAAGAAAT |
| P53 cds for | TTATAGCAATGCACCGACGC |
| P53 cds rev | GACGAACGCCAGCTCAATAG |
| Him/Her for | CGAACCGAGTTGTGGGAAAT |
| Him/Her rev | CCCTTGGAGTGACAATTAGCTG |

DOI: https://doi.org/10.7554/eLife.41637.023

were combined with *UAS-Mi-2 RNAi* or *UAS-w RNAi* (control) and crossed to *tub-Gal80ts; grhNBGal4*. The progeny were kept at 18°C for 7 days and transferred to 30°C for 24 (, Pros) or 48 (Zfh1) hr prior to dissection.

Flies for assessing changes in NB asymmetric mode of division were of the following genotypes:
Control: *UAS-LacZ; Ase-mcherryPonLD*, (*Derivery et al., 2015*)
Overexpressing active Notch: *UAS-NΔecd; Ase-mcherryPonLD*

These flies were crossed to *tub-Gal80ts; grh-Gal4* flies. Crosses were kept at 18°C for 7 days, then shifted to 30°C for 24 hr prior to dissection.

Flies with a CRISPR engineered insertion of GFP into the *zfh1* locus, (Zfh1-GFP (*Albert et al., 2018*) were used to visualise the expression of Zfh1.

## Immunofluorescence

Fixation and immunohistochemistry of larval tissues were performed according to standard protocols. Primary antibodies were guinea pig anti-Dpn (1:2000) courtesy of Christos Delidakis; mouse anti-Mira [1:100, (*Ohshiro et al., 2000*)]; mouse anti-Pros MR1A (1:50, DHSB); rabbit anti-Ase (1:5000, courtesy of Y.N.Jan); rabbit anti-Grh (1:2000,gift of Christos Samakovlis); rabbit anti-Mi-2 (1:10,000, courtesy of Alexander Brehm (*Kreher et al., 2017*). Mouse, rabbit or guinea pig

secondary antibodies were conjugated to Alexa 488, 555, 568, 633 or 647 (Molecular Probes) or to FiTC, Cy3 or Cy5 (Jackson ImmunoResearch). Samples were imaged on a Leica TCS SP8 confocal microscope (Cambridge Advanced Imaging Center (CAIC), University of Cambridge).

## Live imaging of NBs

Larval brains were dissected in dissection media (1.4M NaCl, 26 mM KCl, 4 mM NaH$_2$PO$_4$, 120 mM NAHCO$_3$ and 50 mM Glucose). Brains were dissociated in collagenase solution [2 mg/ml (Sigma, C0130)] for 15 min, rinsed with culture media [Schneider medium (GIBCO 21720–024), 1 mg/ml Glucose (D-L- glucose monohydrate) with 10% FBS (F4135), 2.5% fly extract, 1 mg/ml human insulin (Sigma, I9278) and 1x Antibiotic Antimycotic (GIBCO, 15240–062)] and transferred into microcentrifuge tubes with 40 μl/brain culture media. Brains were subsequently sheared mechanically and the emerging dissociated cells were placed into Poly-D-Lysine coated plates. Cells were left to rest for 30 min prior to imaging for 10–14 hr (*Pampalona et al., 2015*) on a Leica TCS SP8 confocal microscope (Cambridge Advanced Imaging Center (CAIC), University of Cambridge).

## Live imaging of whole brain explants

Whole brains were dissected from 3$^{rd}$ instar larvae in Schneider's medium and cultured according to an improved protocol for long term culturing and imaging of larval brains (*Cabernard and Doe, 2013*; *Hailstone et al., 2017*). Live imaging was performed for 26 hr in an inverted Olympus FV1200 confocal microscope (Ilan Davis Group, Department of Biochemistry, University of Oxford).

## Cell tracking and time-lapse movie image analysis

Cells were segmented in 3D from the his2Av-RFP signal using a combination of median filtering, thresholding, active contour segmentation and 3D watershed to separate nuclei in contact. They were then tracked over time by finding the closest neighbour in a 12px radius and allowing to search in the five previous frames. The mean intensity in the green channel (from E(spl)mγ-GFP) was obtained from the overlap with the tracked nuclei. Cells that were not correctly tracked were manually corrected afterwards and mother-daughter cells assigned. To account for possible differences in fluorescence levels between movies, mean intensity values were normalised to an average of the whole movie. The tracking and posterior analysis codes were implemented in MATLAB (2018a, Mathworks) and R respectively and the scripts are available at https://github.com/juliafs93/Cell-Tracker (*Falo Sanjuan, 2019*; copy archived at https://github.com/elifesciences-publications/CellTracker).

## Cell culture

Drosophila Kc 167 cell line was used. They were obtained from the Drosophila Genomics Resource Centre (https://dgrc.bio.indiana.edu/Home), the community repository for Drosophila cell lines. These cells are not susceptible to Mycoplasma. Kc cells were cultured at 25°C in Schneider's media (GIBCO S0146), supplemented with 5% FBS (Sigma, F9665) and 1x Antibiotic Antimycotic (GIBCO, 15240–062). For Notch activation, Kc cells were treated with 4 mM EGTA in PBS for 30 min. EGTA destabilises the Notch negative regulatory region, exposing the site for ADAM proteases which renders the reminaig transmembrane fragment a substrate for γ-secretase cleavage and release of NICD (*Gupta-Rossi et al., 2001*; *Ilagan et al., 2011*; *Krejcí and Bray, 2007*).

## Mi-2 RNAi in Kc cells

dsRNA for Mi-2, H or GFP (control) was prepared as following: dsRNA was transcribed with MEGA-script T7 Transcription Kit (AMB 13345) using 500 bp PCR fragments specific for each gene flanked by T7 promoter sequences. The newly synthesised dsRNA was treated with DNase, purified with phenol chloroform extraction, precipitated with isopropanol, re-suspended in DEPC water and annealed at 45°C for 1 hr and 15 min. dsRNA was subsequently stored at −20°C.

For single *Mi-2 RNAi*, *H RNAi* or *GFP RNAi* treatment: In a six-well plate, the medium from Kc cells ($2 \times 10^6$ cells per well) was replaced with 20 μg dsRNA diluted in 600 μl Optimem media [Gibco/Life technologies (now part of Thermo-Fisher), 51985–026] for 30 min and 1.5 ml of tissue culture media were subsequently added according to standard protocols. Changes in RNA levels were assessed 72 hr later. For double knockdown, Kc cell medium was replaced with 10 μg dsRNA of

each gene diluted in 600 µl Optimem media for 30 min. 1.5 ml of tissue culture media were subsequently added according to standard protocols. mRNA levels were measured again after 72 hr.

## RNA isolation and qPCR

Kc cells were harvested in Tri reagent and incubated for 10 min. RNA was extracted using phenol chloroform and precipitated in isopropanol overnight at $-20°C$. RNA was resuspended in DEPC treated water and treated with DNAse (Ambion,DNA-free kit, AM1906) to remove genomic DNA. The equivalent of 2 µg of RNA was reverse transcribed with random primers [Oligo(dT)15 Primers (Promega C1101)] using M-MLV reverse transcriptase (Promega M531A). The cDNA products were subsequently diluted 1:5 and 1 µl was used as a template in each quantitative PCR reaction. Quantitative PCR was performed using LightCycler 480 SYBR Green I Master PCR Kit (Roche 04707516001) with a Roche Light Cycler. Samples were normalised to Rpl32. All primers are listed in *Table 1*.

## Chromatin immunoprecipitation

ChIP experiments were performed as previously described (*Krejcí and Bray, 2007*; *Skalska et al., 2015*) except for Mi-2 ChIP where a combination of 1% formaldehyde with 1mmM EGS (ethylene glycol bis(succinimidyl succinate)) was used for cross-linking, (15 min at room temperature). A plate with $15 \times 10^6$ cells was used as a starting material for each ChIP and the following antibodies were used: goat α-Su(H) (Santa Cruz Biotechnology sc15813; 12 µl), rabbit α-H3K56ac (Active Motif 39281; 3 µl), rabbit α-H3K27ac (Abcam ab4729; 10 µl from 0.1 µg/µl), rabbit α-H3K27me3 (Millipore, 07–449; 1 µl), rabbit α-H3 (Abcam ab1791; 1 µl), rabbit-a-Mi-2, (courtesy of Alexander Brehm; 2 µl). Regions of enrichment were analyzed by qPCR using the primers in *Table 2*.

## Co-immunoprecipitation and western blot

For detecting protein levels, control, Mi-2RNAi and H-RNAi-treated Kc cells (~$8 \times 10^6$ cells) were harvested and incubated in Lysis Buffer (50 mM Tris pH7.5, 150 mM NaCl, 10% glycerol, 0.5% triton X-100, Complete protease inhibitors (Roche)) on ice for 30 min. Upon centrifugation, cell debris was pelleted and the supernatant with the nuclear protein extracts was collected and denatured in SDS-sample buffer (100 mM Tris-Cl, pH6.8, 20% Glycerol, 4% SDS, 0.025% Bromophenol blue, 2% β-mercaptoethanol).

For co-immunoprecipitations, $15 \times 10^6$ cells from *Hairless-GFP* or a *control GFP*-expressing stable Kc cell line were incubated in IP Lysis buffer (50 mM Tris-HCL pH8, 150 mM NaCl, 10% Glycerol, 0.5% Triton X-100, and proteinase inhibitor cocktail) on ice for 40 min. Cell debris was pelleted via centrifugation and the supernatants were incubated with 100 µl of Protein G agarose beads (Roche) for 1 hr at 4°C for preclearing. Rabbit anti-GFP (1:1.000, Invitrogen A11122) was then added to the protein extracts and incubated overnight at 4°C. 60 µl of protein G-Agarose beads were added and incubated with the protein extracts for 2 hr at 4°C before washing five times in IP buffer. The pelleted beads were then resuspended in SDS-sample buffer.

Total denatured proteins or immunoprecipitated proteins were resolved by 7% SDS–polyacrylamide gel electrophoresis (BioRad) and transferred to nitrocellulose membrane. Membranes were blocked in TBTM (TBS, 0.05% tween, 3% milk) for 1 hr prior to an overnight incubation at 4°C with primary antibodies (Rabbit anti-GFP 1:2000, Invitrogen A11122; Rabbit anti-Su(H) 1:400 Santa Cruz sc-28713; Rabbit anti-Mi-2 1:10,000 courtesy of A. Brehm; rat anti-tubulin 1:2000). Following three washes of 15 min at RT in TBT (TBS +0.05 Tween), membranes were incubated with horseradish peroxidase-conjugated secondary antibodies (1:2000, HRP Goat anti-rabbit or 1:2000, HRP Goat anti-rat) for 1 hr at RT then washed 3 times for 15 min in TBT. Bound antibodies were detected by the Amersham ECL detection system (GE Healthcare Life Sciences) and documented on X-Ray film.

## Acknowledgements

We are grateful to Judith Pampalona for introducing us to the methods for NB cultures, to Martin Hailstone, Lu Yang and Ilan Davis for help with whole brain long term cultures and to Zoe Pillidge for advice on the molecular experiments in Kc cells. We acknowledge the Bloomington Stock Center, the VDRC Stock Center and the Developmental Studies Hybridoma Bank for Drosophila strains and antibodies and we thank Christos Delidakis and Alexander Brehm for antibodies. We appreciate the many valuable discussions with other members of the Bray lab during this project and we thank

Torcato Martins, Jonty Townson and Kat Millen for critical reading of the manuscript. This work was funded by a program grant from the MRC to SJB and by studentship from the Wellcome Trust for JF-S.

## Additional information

### Funding

| Funder | Grant reference number | Author |
|---|---|---|
| Medical Research Council | MR/L007177/1 | Evanthia Zacharioudaki Sarah Bray |
| Wellcome Trust | 109144/Z/15/Z | Julia Falo Sanjuan |

The funders had no role in study design, data collection and interpretation, or the decision to submit the work for publication.

### Author contributions

Evanthia Zacharioudaki, Conceptualization, Formal analysis, Validation, Investigation, Visualization, Writing—original draft, Writing—review and editing; Julia Falo Sanjuan, Software, Formal analysis, Validation, Visualization; Sarah Bray, Conceptualization, Supervision, Funding acquisition, Writing—original draft, Project administration, Writing—review and editing

### Author ORCIDs

Julia Falo Sanjuan (iD) http://orcid.org/0000-0002-3563-4789
Sarah Bray (iD) https://orcid.org/0000-0002-1642-599X

### Decision letter and Author response

Decision letter https://doi.org/10.7554/eLife.41637.027
Author response https://doi.org/10.7554/eLife.41637.028

## Additional files

### Supplementary files

• Transparent reporting form
DOI: https://doi.org/10.7554/eLife.41637.025

### Data availability

All data generated or analysed during this study are included in the manuscript and supporting files. Examples of movies have been provided for Figures 2 and 4.

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
