## [Decision Letter]

Thank you for submitting your article "Mi-2/NuRD complex protects stem cell progeny from mitogenic Notch signalling" for consideration by *eLife*. Your article has been reviewed by three peer reviewers, including Michael Buszczak as the Reviewing Editor and Reviewer #1, and the evaluation has been overseen K VijayRaghavan as the Senior Editor. The following individual involved in review of your submission has also agreed to reveal their identity: Hongyan Wang (Reviewer #3).

The reviewers have discussed the reviews with one another and the Reviewing Editor has drafted this decision to help you prepare a revised submission.

Summary:

In this manuscript, Bray and colleagues show that ectopic activation of Notch signaling results in NB progeny reverting back to a stem cell-like state after a brief period of time using a number of assays including live cell imaging. Loss of Mi-2 and other components of the NuRD complex enhance hyperplasia in response to activated Notch. To investigate how Mi-2 could be regulating cell fate choice, the authors examined how loss of Mi-2 influenced the expression of Notch target genes. Depletion of Mi-2 alone had no discernible effect on *E(spl)mγ-GFP* in GMCs. However, in the context of activated Notch, they observed a striking persistence of *E(spl)mγ-GFP* in the GMCs in a substantial proportion of lineages. Further examination revealed that mRNA levels of several Notch target genes increase in the absence of MI-2. To investigate the mechanisms by which loss of Mi-2 influences the expression of these Notch target genes, the authors carried out co-IPs and chromatin immunoprecipitation. Work performed using Kc cells suggest Mi-2 does not physically interact with Hairless. In Kc cells, Su(H) recruitment to specific gene promoters in the presence or absence of Notch activation is not altered by knockdown of Mi-2. While these data are mostly negative, the authors do go on to show that Mi-2 does cooperate with Zhf-1 to repress Notch target genes.

The paper provides novel insights into the mechanism by which Mi-2 and the NuRD complex influence Notch target gene expression in NB progeny. In general, the data are nicely presented and the story will likely be of broad interest to the fields of neurodevelopment and stem cell biology.

Essential revisions:

The reviewers raise a number of concerns that must be adequately addressed before the paper can be accepted. Some of the required revisions will likely require further experimentation within the framework of the presented studies and techniques.

1) The hyperplasia phenotype may require more than the ectopic activation of *E(spl)* genes. It is important to extend the observations to other Notch target genes. One way to address this would be to compare the list of Notch targets genes in Kc cells with expression profiling produced upon knock-down of Mi-2.

2) The authors should provide better evidence of direct recruitment of Mi-2/NuRD to Notch target genes in Kc cells.

3) The authors should test whether *zfh1* loss also enhances Nact hyperplasia and whether *zfh1* suppresses the ability of NB progeny to respond to Notch? If they do, these results will strengthen the link between Zfh1 and Notch signaling in NB progeny.

4) The intermediate size of the ectopic NBs observed upon Nact expression may arise from symmetric division of pre-existing NBs. To exclude this possibility, asymmetric division of NBs needs to be examined in Nact VNC NBs.

---

## [Author Response]

The reviewers raise a number of concerns that must be adequately addressed before the paper can be accepted. Some of the required revisions will likely require further experimentation within the framework of the presented studies and techniques.1) The hyperplasia phenotype may require more than the ectopic activation of E(spl) genes. It is important to extend the observations to other Notch target genes. One way to address this would be to compare the list of Notch targets genes in Kc cells with expression profiling produced upon knock-down of Mi-2.

We agree with the reviewers that it is important to extend our findings to other Notch regulated enhancers although we note that ectopic *E(spl)mγ* expression alone is sufficient to recapitulate many of the effects produced by NICD in NB lineages (Zacharioudaki et al., 2012). We have therefore taken advantage of our previous data identifying Notch target genes in the larval brain (Zacharioudaki et al., 2016) and Kc cells (Skalska et al., 2015) and analyzed the effects of Mi2 depletion on several of these characterized targets. First, using previously characterized reporters or antibody staining we show that the Notch targets *pathetic, cyclin E, grainy head* and *miranda* are all up-regulated in NB-like cells when Mi-2 is knocked down in Nact NB lineages. These data are presented in new Figure 3—figure supplement 2.

Second, we analyzed four other Notch target genes in Kc cells, *CG12290 CG17119, Notch* and *rgl (Ral guanine nucleotide dissociation stimulator-like)* and found that all are expressed at higher levels in Mi-2 depleted cells, in both Notch OFF and Notch ON conditions. These new data are added in new Figure 5—figure supplement 1B.In addition we have, as suggested, utilized our genome-wide profiles from Kc cells and compared the Su(H) binding profiles with those of Mi-2 in the same cell-type, (modENCODE data). Strikingly, 72.34% of the 376 Su(H) bound regions overlap with regions bound by dMi-2_in Kc cells, indicating a widespread relationship. This information is added to the Results subsection “Mi-2 loss leads to de-repression of *E(spl)*-C genes”.

2) The authors should provide better evidence of direct recruitment of Mi-2/NuRD to Notch target genes in Kc cells.

We thank the reviewers for this valuable suggestion and we have now included new data to demonstrate that Mi-2 is directly recruited at Notch regulated enhancers. First, as discussed above, we have been able to take advantage of existing ModENCODE data profiling the genomic occupancy of Mi-2 in Kc cells as well as a recent data-set from S2 cells. Both these data show clear enrichments for Mi-2 at the *E(spl)-C* locus in regions that correlate with Su(H) bound enhancers. These graphs have been added to Figure 5A.

Second, a global comparison shows that 72.34% of Su(H) bound regions overlap with Mi-2, indicating that Mi-2 is generally present at Notch regulated enhancers. We have added a statement to this effect in the Results subsection “Mi-2 loss leads to de-repression of *E(spl)*-C genes” Finally we have been able to validate the binding of Mi-2 at *E(spl)-C* and in the vicinity of *CG17119* by using a modified protocol for chromatin immunoprecipitation, These data are now presented in new Figure 5H.

3) The authors should test whether zfh1 loss also enhances Nact hyperplasia and whether zfh1 suppresses the ability of NB progeny to respond to Notch? If they do, these results will strengthen the link between Zfh1 and Notch signaling in NB progeny.

We thank the reviewers for these suggestions. We note that we had already presented results showing that knockdown of *zfh1* led to enhancement of >Nact NB hyperplasia (Figure 6A). We have now added further experiments to strengthen the link between Zfh1 and Notch by showing that ectopic expression of Zfh1 in Nact NB lineages leads to a reduction in the Notch induced hyperplasia. Furthermore, this effect was partially cancelled out by concomitant knock down of Mi-2. These data are now presented in new Figure 6—figure supplement 2A, B.

4) The intermediate size of the ectopic NBs observed upon Nact expression may arise from symmetric division of pre-existing NBs. To exclude this possibility, asymmetric division of NBs needs to be examined in Nact VNC NBs.

We appreciate the concerns of the reviewer. However, we note that the live-imaging of dividing NBs clearly shows that, in Nact conditions, the pre-existing NBs continue to divide asymmetrically both in terms of cell size and inheritance of stem cell markers. Furthermore, when NB-like cells emerge they gradually increase in size then acquire *E(spl)mγ-GFP* before performing their first asymmetric cell division (see P3.1 in Figure 2B and D). To further exclude the possibility that NBs divide symmetrically in Nact conditions we have performed two further experiments. First, we have examined the behavior of Mira and Pon, basally localized molecules during asymmetrical division, with immunofluorescent staining in control vs. Nact NB lineages. Both proteins show normal localization and Mira is clearly segregated into the daughter cell even in the Nact lineages (new Figure 1—figure supplement 2A).

Second we have tracked dividing NBs in whole brain preparations from Nact conditions and find in 100% of cases the NB division produces a small GMC progeny and the NB retains its normal size (new Figure 1—figure supplement 2B). All of these data demonstrate that NBs retain their normal asymmetrical divisions even in Nact conditions.